# Fundamental Improvement of Creep Resistance of New-Generation Nano-Oxide Strengthened Alloys via Hot Rotary Swaging Consolidation

**DOI:** 10.3390/ma13225217

**Published:** 2020-11-18

**Authors:** Jiří Svoboda, Lenka Kunčická, Natália Luptáková, Adam Weiser, Petr Dymáček

**Affiliations:** 1Institute of Physics of Materials, Czech Academy of Sciences, Žižkova 22, 616 62 Brno, Czech Republic; kuncicka@ipm.cz (L.K.); luptakova@ipm.cz (N.L.); weiser@ipm.cz (A.W.); pdymacek@ipm.cz (P.D.); 2Faculty of Materials Science and Technology, VŠB–Technical University of Ostrava, 17. Listopadu 15, 708 33 Ostrava, Czech Republic

**Keywords:** oxide dispersion strengthened (ODS) alloy, mechanical alloying, powder hot consolidation, rolling, rotary swaging

## Abstract

New-generation oxide dispersion-strengthened (ODS) alloys with a high volume fraction of nano-oxides of 5% are intended to become the leading creep- and oxidation-resistant alloys for applications at 1100–1300 °C. Hot consolidation of mechanically alloyed powders by intensive plastic deformation followed by heat treatment of the alloys are the key aspects for achieving top creep properties, typically ensured by a coarse-grained microstructure strengthened with homogeneously dispersed, very stable yttrium nano-oxides. The rotary swaging method proves to be favourable for hot consolidation of the new-generation ODS alloy presented. Compared to specimens consolidated by hot rolling, consolidation by hot rotary swaging predetermines the formation of coarse grains with a very high aspect ratio during subsequent secondary recrystallization. Such a grain morphology increases the creep strength of the new-generation ODS alloy considerably.

## 1. Introduction

The development of advanced materials with excellent high-temperature creep and oxidation resistance is one of the most challenging goals of contemporary material research. Ni-based superalloy single crystals [1] applicable up to 1100 °C, oxide dispersion-strengthened (ODS) ferritic alloys [2] applicable up to 1300 °C, and tungsten heavy alloys (THAs) [3] applicable up to 1500 °C, are the top-level metallic materials with the best creep performance.

Ni-based superalloy single crystals are typically strengthened by about 70% volume fraction of cuboidal γ´-precipitates separated by thin channels of disordered γ matrix [4,5]. Sufficient content of Al in the superalloys ensures their excellent oxidation resistance. However, cutting the γ´-phase by dislocation, and instability of the γ´-precipitates due to their coarsening or rafting at temperatures above 900 °C, significantly limits long-term applicability of Ni-based superalloys in the temperature range of 900–1100 °C. The ODS alloys are strengthened by nano-dispersion of very stable Y-based oxides, typically of 5–20 nm in size and 0.5% volume fraction [6,7]. The ferritic matrix of the ODS alloys also allows sufficient alloying by Al (up to 10 wt.%) ensuring their excellent oxidation resistance [8]. Both the very good creep and excellent oxidation resistance are already covered by the existing ODS alloys for temperatures up to 1300 °C. The THAs profit from very high melting point combined with solid solution and deformation strengthening [9,10]. The THAs are, however, disqualified by their poor oxidation resistance and high specific mass. Thus, the most promising candidate for the top creep- and oxidation-resistant alloys for long-term applications in the temperature range of 1100–1300 °C stems from the family of the ODS alloys.

ODS alloys are typically produced in two steps. Homogeneous powder consisting of the matrix and nano-sized Y_2_O_3_ is produced by (i) mechanical alloying (MA) and (ii) consolidated via (a) hot extrusion (HE), (b) hot isostatic pressing (HIP), (c) spark plasma sintering, or their combination [11,12,13,14,15,16,17]. Very recent research by the authors devoted to the development of new-generation ODS alloys with a high volume fraction of oxides of about 5% (being by one order of magnitude higher than in classical ODS alloys) consolidated via hot rolling [6,18] indicates that the consolidation conditions are important factors influencing the final creep properties of the alloys. Nevertheless, additional deformation (thermomechanical) processing of the ODS alloys is often applied to ensure elimination of porosity and enhance their performance. The processing can be performed via numerous conventional, as well as unconventional, forming methods. Among the conventional ones are, for example, rolling and forging. Recently, Auger et al. [19] used hot-cross rolling to improve resistance of 14YWT ODS alloy against radiation, and Zhang et al. [20] applied hot rolling to increase the strength of Fe–9Cr-0.06C-1.5W-0.5Ti-0.18Si-0.35Y_2_O_3_ ODS alloy (wt.% are used in all notations). Kumar et al. [21] studied mechanical properties of Fe-18Cr-2W-0.2Ti-xY_2_O_3_ alloy consolidated from mechanically alloyed powder by forging. Zhou et al. [22] applied combinations of forging, hot rolling, cold rolling, and annealing to ODS310 alloy with Mo addition. However, inhomogeneous bimodal grain size distribution due to incomplete secondary recrystallization and brittle cracking along the grain boundaries are still issues for the ODS alloys [23]. Although coarse grain microstructure achieved by full secondary recrystallization seems to be the key factor for acquiring excellent creep strength, several ODS alloys investigated in the available literature do not meet this requirement. Our experiments indicate that the ultra fine grained microstructure or incomplete secondary recrystallization in the ODS alloy cause the creep strength to reach only a fraction of the creep strength of the completely recrystallized ones. Moreover, in several papers, the mechanical properties of very expensive ODS alloys are studied at loading conditions, for which much better and cheaper substitutes exist. It is highly desirable to limit the studies of ODS alloys to conditions, at which their properties are really exceptional.

The overall performance of the ODS alloys can be improved in two ways, the first of which is modification of their chemical composition. The new-generation ODS alloys are prepared from powders mechanically alloyed up to a degree of homogeneity at which the overall amount of oxygen originating from the input yttria powder and oxidized surfaces of the input metallic powders is completely dissolved and trapped at defects, such as dislocations and vacancies. After canning, the powder is hot rolled under optimized conditions ensuring the elimination of porosity, as well as provoking dynamic recrystallization resulting in ultra-fine grained (UFG) microstructure. Nano-oxides of the typical size of 5 nm, precipitated during the hot consolidation, stabilize the UFG microstructure up to very high temperatures of 1000–1100 °C. If the energy stored during the hot rolling reaches the critical value, complete secondary recrystallization occurs during subsequent annealing at the temperatures of 1200 °C for several hours [6,18]. As the result, the final microstructure consists of coarse grains of the typical size in the order of 100 µm strengthened by homogeneous dispersion of yttrium nano-oxides of the typical size of 20 nm [24]. Tensile tests performed at temperatures exceeding 1000 °C lead to a rather brittle intergranular fracture, which indicates that high-angle grain boundaries are the weakest link in the new-generation ODS alloys due to their limited cohesion strength. However, for creep experiments at the applied stress of 40–60% of the strength measured by the tensile test, the cohesion of high-angle grain boundaries is sufficient and the stationary creep rate drops to the values of the order of 10^−9^ s^−1^, as the dispersion of nano-oxides very effectively suppresses the dislocation creep [25] and the time to fracture reaches hundreds or thousands of hours. Moreover, as the primary creep strain is rather small, typically 0.1–0.2%, the new-generation ODS alloys are advantageous to design components requiring long-term rigidity at very high temperatures.

The second way in which the performance of the ODS alloys can be enhanced is the application of optimized deformation processing (thermomechanical treatment), which can advantageously be performed by unconventional forming techniques and methods of severe plastic deformation (SPD). SPD methods such as equal channel angular pressing (ECAP) [26,27] and related methods (non-ECAP [28], twist channel (multi-) angular pressing [29,30], etc.), or high pressure torsion (HPT) [31] are favourable for processing of small volumes of materials, while methods such as accumulative roll bonding (ARB) [32,33] and rotary swaging (RS) [34] are advantageous for large bulk products. Rotary swaging (RS) has been used to perform thermomechanical processing of various alloys, from brittle Mg-based compounds [35,36], through bio-applicable NiTi shape memory alloys [37,38], Ti-based materials with enhanced performance [39], and metallic composites [40,41], to FeCo magnetic compounds [42], and modern steels [43]. RS was also applied to the Eurofer’97 type of ODS after consolidation by HIP or HE [44]. RS seems to be very promising particularly for consolidation of canned ODS powders due to its incremental character, prevailing compressive stress state, and high imposed shear strains [45,46,47]. Compared to HE, RS is a standard industrial technology allowing relatively easy and cheap production of solid, hollow, or profiled bars within a wide range of diameters.

The present study introduces the Fe-10Al-3Y_2_O_3_-1Ti new-generation ODS alloy consolidated from a mechanically alloyed powder by two methods of intensive plastic deformation: (i) hot rolling, and (ii) hot RS. To the best of the authors’ knowledge, this is the very first experiment in which hot RS has been utilized for ODS alloy consolidation. To demonstrate the favourable effects of RS, microstructures and creep properties of rolled and swaged samples are compared and the results of experiments are discussed in detail.

## 2. Materials and Methods

Chemically homogeneous Fe-10Al-3Y_2_O_3_-1Ti powder was prepared from Fe, Al, Y_2_O_3_, and Ti powders of 99.9% purity by MA using a self-made ball mill. A vacuum-tight milling container with the volume of 22 dm^3^ and diameter of 400 mm made from low alloyed steel was filled with 100 Fe-1Cr-1C bearing balls of 40 mm diameter (altogether 25 kg). The total amount of 1 kg of the powder was mechanically alloyed by rotation of the milling container along the horizontal axis (70 rpm). After a sufficiently long MA (two weeks) in vacuum, the powder properties became saturated and the powder particles consisted of a homogeneous solid solution with a huge density of defects, like dislocations and vacancies (see [6] for more details). The rather wide size distribution (between 2 to 200 μm) of the cold compacted powder can be determined from Figure 1. It is very complicated to determine the mean particle size due to very wide size distribution. Moreover, the powder involves a huge density of defects and the grains and their grain boundaries are not well defined. The measured content of Cr (0.05%) in the MA powder indicates that about 5% of the milling balls is introduced into the MA powder by the abrasion of milling balls and thus the MA powder contains also 0.05% of C. The influence of milling media on the microstructure and mechanical properties of mechanically milled and sintered aluminium was analyzed e.g. in [48].

The MA powder was hot consolidated by using two techniques: (i) rolling and (ii) RS. The rolling cylinders of diameter 75 mm in the mill rotated at the speed of 65 rpm. The rotary swaging machine by the HMP company allowed stepwise reduction of axisymmetric bars. Four swaging dies performed high-frequency radial movements with short strokes and applied compressive force to the inserted workpiece. The forming process took place in many small processing steps and, therefore, RS can be described as an incremental forming process. Infeed swaging with fleeting clamping of the swaged sample was used. The entire swaging was performed at the temperature of 950 °C with no lubrication. The specimens were heated and kept 5 min at the required temperature in the atmospheric furnace before each consolidation step.

The containers for hot rolling and hot RS made from stainless steel tubes were filled with the mechanically alloyed powder, evacuated, and sealed by welding. The rolling container was hot rolled in three steps from the initial diameter of 20 mm to the final thickness of the sheet of 3.2 mm at 960 °C. The swaging container was hot rotary swaged from the initial diameter of 50 mm to the final diameter of 15 mm at the same temperature, to provide similar hot consolidation conditions. The schematic depiction of the rolling and swaging processes of the new-generation ODS alloy, including geometries of the canned semi-products and definition of the further described directions, is shown in Figure 2. After stripping from the containers, the consolidated new-generation ODS alloys were secondarily recrystallized by annealing in a vacuum furnace at 1200 °C for 4 h.

The study deals with two sets of specimens of the secondary recrystallized new-generation ODS alloy. Flat specimens with the gauge length of 25 mm and cross section of 2.5 mm × 3.5 mm cut by precise water jet and grinded from the rolled sheets are denoted as Specimens I, while circular specimens with the gauge length of 25 mm and diameter of 6 mm produced from the swaged rods are denoted as Specimens II. The tensile and creep tests were performed using Zwick/Roell—Messphysik KAPPA LA (Fürstenfeld, Austria) spring 20 kN creep test system equipped by MAYTEC furnace (Singen, Germany) with the working temperatures up to 1400 °C. One specimen for a tensile test at constant rate 10^−6^ s^−1^ and four, respectively five, specimens were used for the creep tests of Specimens I and II (i.e., one specimen for each applied stress). The microstructures were observed using a scanning electron microscope (SEM) Tescan Lyra 3 XMU FEG/SEMxFIB (Tescan, Brno, Czech Republic) equipped with X-Max80 EDS (energy-dispersive X-ray) detector for X-ray microanalysis and EBSD (electron backscatter diffraction) detector with Aztec control system (Oxford Instruments, Abingdon, UK).

## 3. Mechanical Testing

The tensile tests performed at the temperature of 1100 °C and strain rate of 10^−6^ s^−1^ showed the strength of 75 MPa and ductility of 1.1% for Specimens I, and strength of 115 MPa and ductility of 1.2% for Specimens II. The rather small ductility and the purely intergranular fracture indicated that the grain boundary cohesion strength was the weakest link in the specimens. This weakest link, however, did not necessarily play an important role in the creep experiments at the applied stress significantly below the measured tensile strength.

The results of creep tests performed at 1100 °C are summarized in Figure 3. To get an idea about the creep properties of the new-generation ODS alloy, the results are compared with the data sheet for the top commercial ODS alloy, MA 956 [49] (only times to rupture are available). Figure 3a obviously shows that the stress exponent of time to rupture is much higher for Specimens II compared to Specimens I. This can be attributed to the significantly increased creep ductility of Specimens II at low stresses. The stress exponents of minimum creep rate shown in Figure 3b are similar for both Specimens I and II. It is evident from the creep tests (see Figure 3a) that the strength of Specimens II is significantly (by a factor of about 1.5 to 2) higher than that of Specimens I and the minimum creep rate is lower by more than two orders of magnitude (see Figure 3b). To explain these facts, SEM analysis was further performed.

## 4. Microstructure Characterization

The microstructures acquired by SEM (back-scattered electron analysis) from central parts of the samples after hot rolling and hot RS are compared in Figure 4a,b, respectively. The grain microstructure of the rolled specimen (Figure 4a) is very fine (100 nm); the grains are more or less equiaxed (aspect ratio near to 1) and exhibit a rather narrow size distribution. By contrast, the grain microstructure after RS (Figure 3b) is significantly coarser (the width of grains 200 nm) and considerably elongated (aspect ratio 2.2) in the direction of the swaging axis (horizontal direction).

The electron backscatter diffraction (EBSD) analyses of grain microstructures of Specimens I and II after secondary recrystallization, which led to a drastic increase in the grain size by nearly four orders of magnitude, are compared in Figure 5. The shapes of the grains in Specimens I slightly resemble pancakes in the rolling plane (horizontal direction in Figure 5a), the grains in Specimens II exhibit shapes with a high aspect ratio along the swaging axis (horizontal direction in Figure 5b). No significant texture formation is detected from the inverse pole figures (see Figure 5c,d).

The microstructures featuring dispersions of nano-oxides inside the secondary recrystallized grains of Specimens I and II are compared in Figure 6, which show that the oxide dispersion in Specimens I is significantly finer and more homogeneous than in Specimens II. The reason for the difference can be found in the comparison of grain microstructures after hot consolidation presented in Figure 4. The grains after hot rolling are very fine (of the order of 100 nm) and the size of the oxides (typically 5 nm, see Figure 1b in reference [50]) is below the resolution of SEM (typically observed by transmission electron microscopy). The oxides at the grain boundaries coarsen faster than those within the grain interiors during annealing and thus the morphology of the hot-rolled microstructure (see Figure 4a) is imprinted in the oxide dispersion after secondary recrystallization. Only the nano-oxides originally situated at the grain boundaries of the UFG microstructure remain as they consume the nano-oxides from interiors of ultra fine grains during annealing. The grain microstructure after hot RS is much coarser (500–1000 nm) than that after hot rolling and the grains are considerably elongated in the direction of swaging axis, see Figure 4b. Moreover, the swaged structure already contains some portion of relatively coarse nano-oxides (20–50 nm). By these reasons, the nano-oxide dispersion in Specimens II, reflecting the hot-swaged grain microstructure, is coarser and less homogeneous and features a wider size distribution. As the nano-oxides are again situated predominantly in locations of the original grain boundaries after RS, the nano-oxides form chains parallel to the swaging axis at the metallographic section in Figure 6b.

## 5. Discussion

The formation of grains with high aspect ratio in Specimens II by secondary recrystallization (Figure 5b) can be explained as follows. Due to the Zener pinning, the grain boundary of the secondary recrystallizing grain migrates easily across the zones of low-density of nano-oxides and remains anchored in the zones with a very high density of nano-oxides. As the nano-oxides are concentrated predominantly at the locations of the original grain boundaries after RS, i.e., predominantly at surfaces parallel to the swaging axis, the effective mobility is significantly lower for the grain boundaries parallel to the swaging axis than for those normal to the swaging axis. During the subsequent secondary recrystallization, the growth rate of the grains aligned with the swaging axis is higher than the growth rate of the grains aligned with the normal to the swaging axis, which explains the formation of grains with high aspect ratio parallel to the swaging axis after secondary recrystallization (see Figure 5b).

When uniaxial stress is applied at very high temperatures, redistribution of stress within the specimen occurs rapidly and a certain part of the applied stress is equilibrated by shear stresses at the grain boundaries parallel to the direction of the applied stress. The higher fraction of grain boundaries parallel to the loading axis exists in the specimen, and more loading is transferred by this redistribution effect. As the fraction of grain boundaries parallel to the loading axis is significantly higher in Specimens II than in Specimens I, and the cohesive strength of the grain boundaries normal to the applied stress is the given limiting factor, the strength of Specimens II is considerably higher than of that of Specimens I.

The grain geometry with high aspect ratio and nano-oxide distribution pattern leads to anisotropy in the mechanical properties and determines high tensile strength along the swaging axis. This may be advantageous for specific uniaxially loaded components, such as pullrods and grips in testing machines working at very high temperatures. Moreover, excellent oxidation resistance of the Fe-10Al-3Y_2_O_3_-1Ti is guaranteed by the high content of Al forming a protective compact alumina surface film by reaction with the air. The protective film may also prohibit reaction of pullrods and grips with tested specimens.

Special care must be taken to avoid significant inhomogeneities in the grain size, as this may cause secondary cracking or delamination, as described in [51,52]. Thus, complete secondary recrystallization is required to proceed in the samples. The respective model for the treatment of dynamic recrystallization during hot consolidation followed by secondary recrystallization is presented in [50]. The model clearly shows that only selected hot consolidation conditions from the processing window may lead to complete secondary recrystallization during subsequent annealing, which is the case of in this study.

Concerning the mechanical properties at 1100 °C, the influence of the consolidation method is substantial. Both the tensile test strength and creep strength are by a factor of about 1.5 to 2 higher for Specimens II than for Specimens I. This is despite the fact that the nano-oxide dispersion in Specimens I is more regular than that in Specimens II, see Figure 6. Summarizing based on the convincing results, the grains in both Specimens I and II are sufficiently strengthened by the dispersion of nano-oxides and the mechanical properties at 1100 °C are, for the given cohesion of the grain boundaries, determined predominantly by the grain geometry.

Today, ODS alloys are not produced on the commercial basis any more, which is most probably due to their rather complicated processing resulting in a very high price and, consequently, low market demand. The processing of new-generation ODS alloys is much less complicated when utilizing standard industrial procedures such as rolling or RS for hot consolidation. The use of the self-made attritor is also very simple and cheap and it allows easy up-scaling. If the chemical composition and hot-consolidation conditions are optimized to ensure the complete secondary recrystallization as a standard, the power to prize ratio of the new-generation ODS alloys will be significantly higher than that of the recent ODS alloys and commercial interest in ODS alloys renewed.

## 6. Conclusions

The conclusions can be summarized as follows:Two batches of the Fe-10Al-3Y_2_O_3_-1Ti new-generation ODS alloys are prepared by hot consolidation of mechanically alloyed powder using (i) rolling and (ii) rotary swaging.Secondary recrystallization of the hot rotary swaging consolidated specimens featuring ultra-fine-grained microstructure leads to rather coarse grains significantly elongated in the direction of swaging axis.Although the microstructure of nano-oxides in the secondary recrystallized grains of the ODS alloy consolidated by hot rolling is significantly more regular and finer than that of the ODS alloy consolidated by hot rotary swaging, the strength of the latter at 1100 °C is by a factor of 1.5 to 2 higher than that of the first one.The creep strength of the present new-generation of ODS alloy at 1100 °C is predominantly determined by the morphology of grain boundaries being the weakest link due to their limited cohesive strength.The creep strength at 1100 °C of the new-generation ODS alloys consolidated by hot rotary swaging exceeds that of top commercial ODS alloy by more than 30%.The hot consolidation of the ODS powder via rotary swaging compared to rolling is proven advantageous for production of the new-generation high-strength ODS alloys for application at temperatures 1100–1300 °C.

## Figures and Tables

**Figure 1 materials-13-05217-f001:**
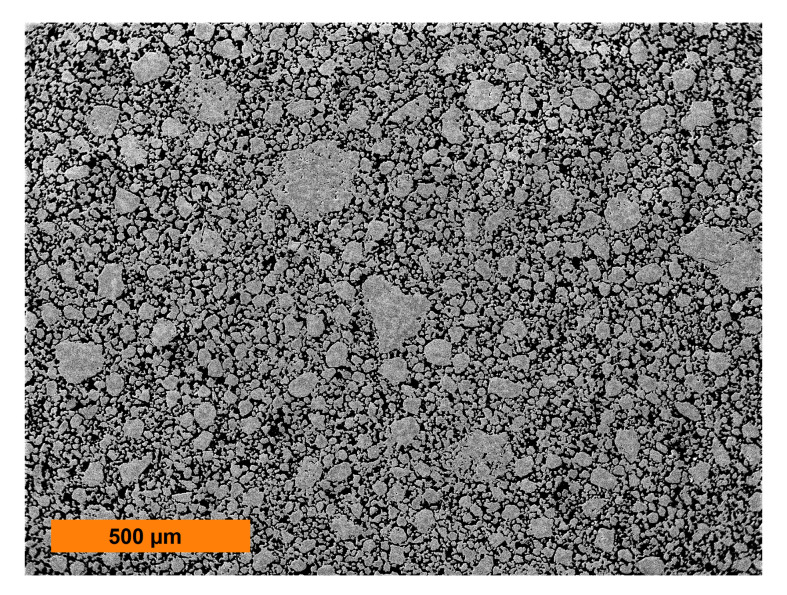
Metallographic section of the mechanically alloyed powder of the new-generation ODS alloy.

**Figure 2 materials-13-05217-f002:**
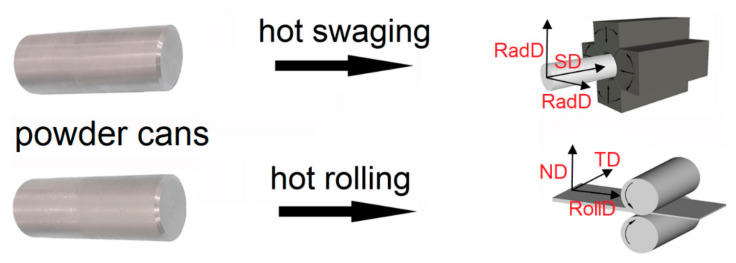
Rolling and swaging geometries used for consolidation of new-generation oxide dispersion-strengthened (ODS) alloys.

**Figure 3 materials-13-05217-f003:**
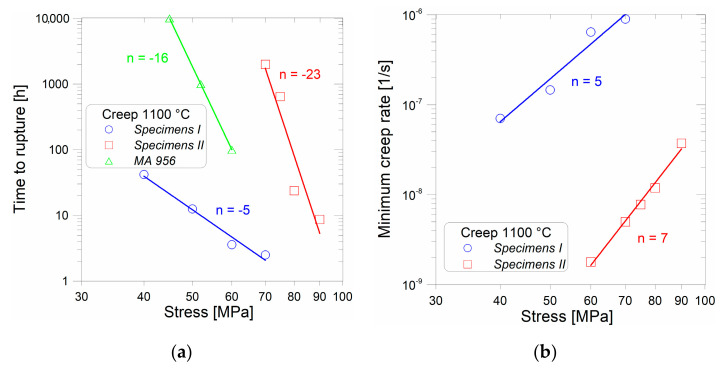
Comparison of (**a**) time to rupture; (**b**) minimum creep rate for Specimens I and II. Interrupted creep test of Specimens II at 60 MPa is used to determine the minimum creep rate.

**Figure 4 materials-13-05217-f004:**
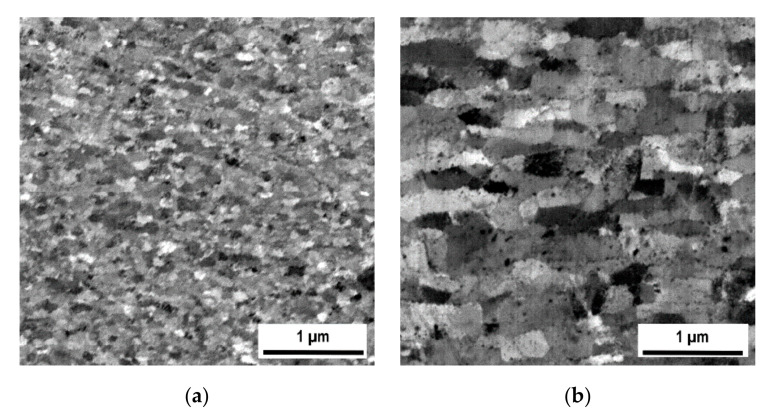
Comparison of microstructures prepared by (**a**) hot rolling in RollDxND plane; (**b**) hot rotary swaging (RS) in SDxRadD plane.

**Figure 5 materials-13-05217-f005:**
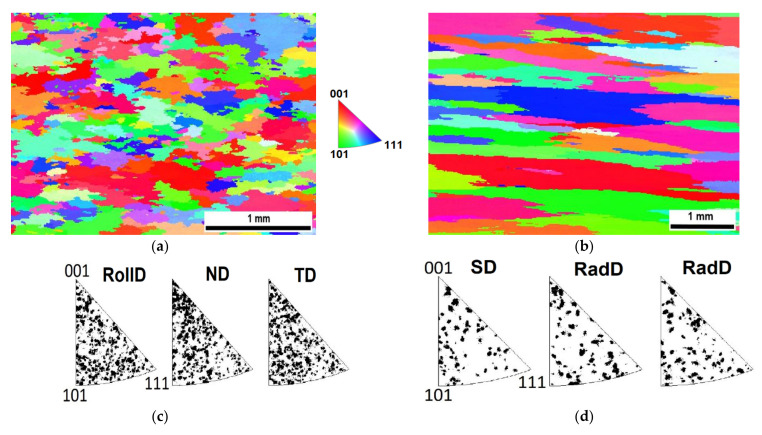
Comparison of grain microstructures after secondary recrystallization (**a**) Specimens I in RollDxND plane; (**b**) Specimens II in SDxRadD plane, scanning electron microscopy (SEM)–electron backscatter diffraction (EBSD) inverse pole figures for (**c**) Specimens I; (**d**) Specimens II. Individual directions are defined in Figure 2.

**Figure 6 materials-13-05217-f006:**
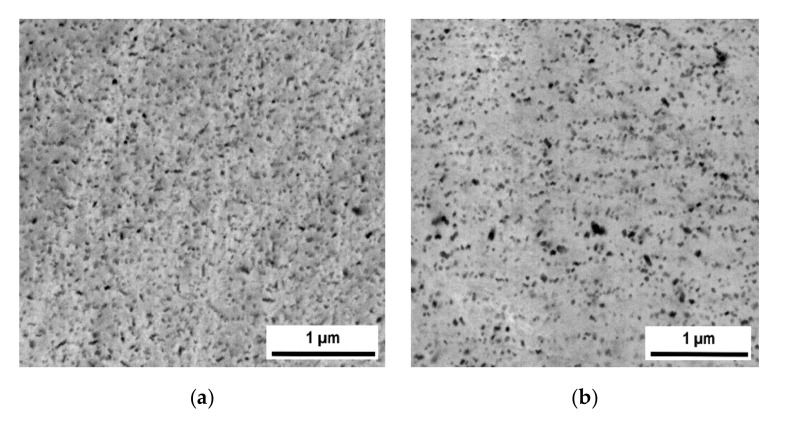
Comparison of microstructures of nano-oxide dispersion within grains of (**a**) Specimens I (**b**) Specimens II (SEM—backscattered electrons).

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
