# Peer review of "Fundamental Improvement of Creep Resistance of New-Generation Nano-Oxide Strengthened Alloys via Hot Rotary Swaging Consolidation"

_materials, 2020, doi:10.3390/ma13225217_

Round 1

Reviewer 1 Report

The manuscript ‘Fundamental improvement of creep resistance of new-generation nano-oxide strengthened alloys via hot rotary swaging consolidation’ is well-organized, with different kind of analyses and interesting.

The authors compared creep strength of the new-generation ODS alloy consolidated by hot rotary swaging versus hot rolling.

I have only few remarks to improve the overall quality of the work; so I suggest to publish it onto Materials after minor revisions.

  1. First of all the title: Authors could use the titile ‘Improvement of creep resistance of new-generation nano-oxide strengthened alloys via hot rotary swaging consolidation’ giving up at ‘fundamental’.
  2. Lines 66-68 ‘Although coarse grain microstructure achieved by full secondary recrystallization seems to be the key factor for acquiring excellent creep strength, several ODS alloys investigated in the available literature do not meet this requirement.’ have to be better explained and argued. Please add references. Most published works found that the activation energies of the creep were higher in case of UFG than CG for the same alloy.
  3. Please rephrase to improve the meaning of lanes 68-69. 'Moreover, mechanical properties of very expensive ODS alloys are studied at loading conditions, for which much better and cheaper substitutes exist'.
  4. Sometimes authors write thermomechanical but they use also thermos-mechanical – please use one of them.
  5. Figure 2 a presents the results compared with the data sheet for the top commercial ODS alloy, MA 956 but in figure 2 b the comparison are missing– please improve.
  6. The authors use ‘needle-like’ for elongated grain in one direction. Perhaps it will be better to use ‘textured’. The needle-like morphology is properly used in case of martensite solid solution.
  7. Line 142: strain rate of 10-6 must be corrected.
  8. Materials and methods: the authors should add a specific paragraph about the rolling mill the rotary swaging machines and applied technologies. Also a short description of RS technology would be useful.

Author Response

We thank to Reviewer 1 for her/his comments that significantly improve the quality of this paper.

The manuscript ‘Fundamental improvement of creep resistance of new-generation nano-oxide strengthened alloys via hot rotary swaging consolidation’ is well-organized, with different kind of analyses and interesting.

The authors compared creep strength of the new-generation ODS alloy consolidated by hot rotary swaging versus hot rolling.

I have only few remarks to improve the overall quality of the work; so I suggest to publish it onto Materials after minor revisions.

  1. First of all the title: Authors could use the title ‘Improvement of creep resistance of new-generation nano-oxide strengthened alloys via hot rotary swaging consolidation’ giving up at ‘fundamental’.

Reply: After some considerations, we decided that the increase in creep properties due to RS is really fundamental (factor 2) and decided to keep the word "fundamental" in the title of the paper. Omitting the word "fundamental" could also reduce the attractivity of the paper and its "selling potential".

  1. Lines 66-68 ‘Although coarse grain microstructure achieved by full secondary recrystallization seems to be the key factor for acquiring excellent creep strength, several ODS alloys investigated in the available literature do not meet this requirement.’ have to be better explained and argued. Please add references. Most published works found that the activation energies of the creep were higher in case of UFG than CG for the same alloy.

Reply: This is in total opposite with our measurements. We determined the creep strength also of the UFG of our ODS alloys being less then 20 % of the CG one. A higher activation energy of creep need not correlate with a higher creep strength. A comment is included into the text. 

  1. Please rephrase to improve the meaning of lanes 68-69. 'Moreover, mechanical properties of very expensive ODS alloys are studied at loading conditions, for which much better and cheaper substitutes exist'.

Reply: A better argumentation is given in this part.

  1. Sometimes authors write thermomechanical but they use also thermo-mechanical – please use one of them.

Reply: Only thermomechanical is used in the text now. 

  1. Figure 2 a presents the results compared with the data sheet for the top commercial ODS alloy, MA 956 but in figure 2 b the comparison are missing– please improve.

Reply: The creep data for MA 956 are taken from the Ref. [48] (now [50]), where the creep rates are not provided.  

  1. The authors use ‘needle-like’ for elongated grain in one direction. Perhaps it will be better to use ‘textured’. The needle-like morphology is properly used in case of martensite solid solution.

Reply: “needle-like” is replaced by “with high aspect ratio”.

  1. Line 142: strain rate of 10-6 must be corrected.

Reply: Corrected.

  1. Materials and methods: the authors should add a specific paragraph about the rolling mill the rotary swaging machines and applied technologies. Also a short description of RS technology would be useful.

Reply: The paragraph is included.

Reviewer 2 Report

This paper proposes a new-generation nano-oxide strengthened alloy Fe-10Al-3Y2O3-1Ti using hot rotary swaging consolidation and investigates the strength and creep properties at 1100 °C, in comparison with those using rolling consolidation. It shows the advantageous behaviour using rotary swaging consolidation for this alloy. Microstructure was examined in order to study the mechanisms. The paper is well written and concise. However, it could be improved if the following issues are addressed.

Major issues:

  • The main concern is the use of terminology “secondary recrystallization”. To my knowledge, secondary recrystallization refers to abnormal grain growth, in which a very small number of grains grow selectively at the expense of many other primary recrystallized grains. But the paper shows fairly uniform grain size after recrystallization.
  • In Abstract (L17) and Conclusions (L269): “1100-1300 °C”. The results reported are only for 1100 °C, therefore the temperature range in the abstract and conclusions is not appropriate. This should be modified. Better still, please present relevant results at 1300 °
  • In results (L192): “The grain microstructure after hot RS is much coarser than that after hot rolling”. Providing quantitative values of the average grain sizes would be helpful.

Minor issues:

  • L43/153/236, “both,” – comma should be removed.
  • L142, “10-6 s-1” – “-1” should be superscript.
  • L207, “predominantly at surfaces parallel to” – surfaces should be changed to grain boundaries.
  • L253, “Two grades of” – they are the same grade but with different processing route.

Author Response

We thank to Reviewer 2 for her/his comments that significantly improve the quality of this paper.

This paper proposes a new-generation nano-oxide strengthened alloy Fe-10Al-3Y2O3-1Ti using hot rotary swaging consolidation and investigates the strength and creep properties at 1100 °C, in comparison with those using rolling consolidation. It shows the advantageous behaviour using rotary swaging consolidation for this alloy. Microstructure was examined in order to study the mechanisms. The paper is well written and concise. However, it could be improved if the following issues are addressed.

Major issues:

  1. The main concern is the use of terminology “secondary recrystallization”. To my knowledge, secondary recrystallization refers to abnormal grain growth, in which a very small number of grains grow selectively at the expense of many other primary recrystallized grains. But the paper shows fairly uniform grain size after recrystallization.

Reply: The mechanism and kinetics of secondary recrystallization is analyzed in [6,17]. The mechanism is exactly that you mention.   

  1. In Abstract (L17) and Conclusions (L269): “1100-1300 °C”. The results reported are only for 1100 °C, therefore the temperature range in the abstract and conclusions is not appropriate. This should be modified. Better still, please present relevant results at 1300 °C.

Reply: The alloys are indeed considered as creep and oxidation resistant alloys for applications at 1100-1300°C, although they were tested only at 1100 °C. The fact that the tests are performed only at 1100 °C is repeatedly explicitly written, a confusion is thus not possible.    

  1. In results (L192): “The grain microstructure after hot RS is much coarser than that after hot rolling”. Providing quantitative values of the average grain sizes would be helpful.

Reply: Included into the text

Minor issues:

L43/153/236, “both,” – comma should be removed.

Reply: Corrected.

L142, “10-6 s-1” – “-1” should be superscript.

Reply: Corrected.

L207, “predominantly at surfaces parallel to” – surfaces should be changed to grain boundaries.

Reply: We must insist on the “surfaces”, because the configuration of the oxides remains kept also after the detaching of the grain boundary from the oxides and its migration away. 

L253, “Two grades of” – they are the same grade but with different processing route.

Reply: Changed to batches.

Reviewer 3 Report

This study focuses on the creep behaviour of an Fe based oxide dispersion strengthened (ODS) alloy. The alloy is prepared by mechanical alloying, and powders are consolidated by hot rotary swaging. Hot rolling consolidation is also employed for comparison purpose.

The methodology followed is adequate, and the document is clearly written. Nevertheless, some questions are required to be clarified before publication is recommended:

  • SEM brand and model must be specified in “Materials and Method” section
  • Lines 117-120. It is not clearly stated if the milling conditions (ball mass, ball to powder ration, rotor speed,…) are exactly the same that the authors used in reference #6.

Also, the milling balls alloy must be specified. The ball material has and very important effect over the properties of milled powder, and some reference, as follows, should be included:

    • Influence of milling media on the microstructure and mechanical properties of mechanically milled and sintered aluminium. Journal of Materials Science, 2005, 40(15).
  • Please, use superscript in lines 142, 154, 236 and 260.
  • For any powder metallurgy process, the characteristics of the powder before its consolidation are key factors.

Please include SEM and/or TEM micrographs of the AM powder, or a reference where they can be seen.

What is the mean particle and grain size of the milled powder?

  • How many samples have been employed to carry out the tensile and creep test?.
  • Please, could you provide more details about the phrase in lines 186-187 in which is stated that the size of oxides after consolidation is 5 nm?.

The oxides mean size is the same using hot rolling or hot RS?

  • Some references, as follows, should be included (at the Introduction section) to point some fast methods used to consolidate ODS alloys:
    • A One-Dimensional Model of the Electrical Resistance Sintering Process. Metallurgical and Materials Transactions A, 2015, 46(2).

Author Response

We thank to Reviewer 3 for her/his comments that significantly improve the quality of this paper.

This study focuses on the creep behaviour of an Fe based oxide dispersion strengthened (ODS) alloy. The alloy is prepared by mechanical alloying, and powders are consolidated by hot rotary swaging. Hot rolling consolidation is also employed for comparison purpose.

The methodology followed is adequate, and the document is clearly written. Nevertheless, some questions are required to be clarified before publication is recommended:

  1. SEM brand and model must be specified in “Materials and Method” section

Reply: Specified.

  1. Lines 117-120. It is not clearly stated if the milling conditions (ball mass, ball to powder ration, rotor speed,…) are exactly the same that the authors used in reference #6.

Also, the milling balls alloy must be specified. The ball material has and very important effect over the properties of milled powder, and some reference, as follows, should be included:

Reply: The actual milling conditions are included.

Influence of milling media on the microstructure and mechanical properties of mechanically milled and sintered aluminium. Journal of Materials Science, 2005, 40(15).

Reply: Reference added.

  1. Please, use superscript in lines 142, 154, 236 and 260.

Reply: Corrected.

  1. For any powder metallurgy process, the characteristics of the powder before its consolidation are key factors.

Reply: We added the powder characteristics.

  1. Please include SEM and/or TEM micrographs of the MA powder, or a reference where they can be seen.

Reply: SEM micrograph and the respective text are added. The TEM analysis is irrelevant because the MA powder contains too many defects and nothing can be distinguished in TEM.

What is the mean particle and grain size of the milled powder?

Reply: This can be estimated from the newly included Figure 1. It is very complicated to determine the mean particle size due to very wide size distribution. Moreover, the powder involves a huge density of defects and the grains are not well defined. Respective comments are included into the text.    

  1. How many samples have been employed to carry out the tensile and creep test?

Reply: The number of specimens used in the study was added to the section 2.

  1. Please, could you provide more details about the phrase in lines 186-187 in which is stated that the size of oxides after consolidation is 5 nm?.

The oxides mean size is the same using hot rolling or hot RS?

Reply: The reference to 5 nm oxides after hot rolling is included. The mean size of oxides after hot RS is estimated from Figure 4b. 

  1. Some references, as follows, should be included (at the Introduction section) to point some fast methods used to consolidate ODS alloys:

A One-Dimensional Model of the Electrical Resistance Sintering Process. Metallurgical and Materials Transactions A, 2015, 46(2).

Reply: Reference added.